**Subject Area:**
cellular biology/molecular biology

RNA-seq, endothelial cells, lipid peroxidation, PPARα, CCL2

**Authors for correspondence:**
Zhihua Yu
e-mail: yzh1109@hotmail.com
Chuan Chen
e-mail: chen9453@126.com

†Fangfang Dou, Beiling Wu and Lin Sun contributed equally to this work.

# Identification of a novel regulatory pathway for PPARα by RNA-seq characterization of the endothelial cell lipid peroxidative injury transcriptome

Fangfang Dou†, Beiling Wu†, Lin Sun†, Jiulin Chen, Te Liu, Zhihua Yu and Chuan Chen

Shanghai Geriatric Institute of Chinese Medicine, Shanghai University of Traditional Chinese Medicine, Shanghai 200031, People's Republic of China

ZY, 0000-0001-5326-5545; CC, 0000-0002-5742-5099

Endothelial dysfunction caused by endothelial cell injuries is the initiating factor for atherosclerosis (AS), and lipid peroxidative injury is one of a dominant factor for AS pathogenesis. Using RNA-seq, we compared changes in transcriptome expression before and after endothelial cell injury, and found 311 differentially expressed genes (DEGs), of which 258 genes were upregulated and 53 genes were downregulated. The protein–protein interactions (PPIs) between the genes were analysed using the STRING database, and a PPI network of DEGs was constructed. The relationship distributions among these PPIs were analysed by performing network node statistics. We found that in the top 20 DEGs with high connected protein nodes in the PPI network, 16 were upregulated and 4 were downregulated. Gene ontology (GO) functional enrichment analysis and KEGG pathway enrichment analysis on the DEGs were also performed. By comparing the upregulated expressed genes with high connected protein nodes in the PPI network to those related to endothelial cell lipid damage and repair in the GO analysis, we identified seven genes (NOX4, PPARA, CCL2, PDGFB, IL8, VWF, CD36) and verified their expression levels by real-time polymerase chain reaction. The protein interactions between the seven genes were then analysed using the STRING database. The results predicted that CCL2 interacts with NOX4, PPARα, PDGFβ and VWF individually. Thus, we examined the protein expression levels of CCL2, NOX4, PPARα, PDGFβ and VWF, and found that the expression levels of all proteins were significantly upregulated after the lipid peroxidative injury, with CCL2 and PPARα exhibiting the highest expression levels. Therefore, we investigated the interregulatory relationship between CCL2 and PPARα and their roles in the repair of endothelial cell injury. With the help of gene overexpression and knockdown techniques, we discovered that PPARα promotes the repair of endothelial cell injury by upregulating CCL2 expression in human umbilical vein endothelial cells but that CCL2 cannot regulate PPARα expression. Therefore, we believe that PPARα participates in the repair of endothelial cell lipid peroxidative injury through regulating the expression of CCL2.

## 1. Introduction

A transcriptome is a collection of all transcripts produced by a species or a specific cell type. Transcriptome studies can improve the overall understanding of the functions and structures of genes and reveal the molecular mechanisms of specific biological processes or the pathogenesis of a disease. Compared with

traditional chip hybridization, transcriptome sequencing (RNA-seq) allows for the detection of the overall transcriptional activity of any species without the requirement of predesigned probes for known sequences, providing more accurate digital signals with high throughput and a broad detection range [1,2]. The development of atherosclerosis (AS) is closely related to endothelial dysfunction caused by endothelial cell injury. Vascular endothelial cells regulate thrombosis, inflammation, vascular tone and vascular remodelling by producing effector molecules. Increases in plasma cholesterol levels promote the oxidation of unused or unmetabolized cholesterol to produce oxycholesterols, thereby causing endothelial cell injury. Injured endothelial cells initiate a series of repair mechanisms to prevent the cell apoptosis induced by excessive oxycholesterols. Therefore, the repair of injured endothelial cells is the first line of defence against AS [3,4].

In this study, we established an *in vitro* vascular endothelial cell injury model featuring endothelial cell lipid peroxidative injury and used RNA-seq and relevant data analysis to identify genes involved in the repair of endothelial cell injury [5]. By comparing the upregulated genes with high connected protein nodes in the protein–protein interaction (PPI) network to those related to endothelial cell lipid damage and repair in the gene ontology (GO) analysis, we identified seven genes and examined whether any could predict PPIs between the seven genes using the STRING database. We discovered that CCL2 is a central gene and has multiple connections with other genes. In the protein expression assays, the CCL2 and PPARα expression levels showed the most significant increases, suggesting that the interactions between CCL2 and PPARα and their effects on the repair of endothelial cell injury should become a focus of our investigation. CCL2 belongs to the chemokine family and is secreted by various cells, such as fibroblasts, vascular smooth muscle cells (VSMCs), endothelial cells and monocytes. Through interactions with its receptors, CCL2 is involved in many physiological functions, such as the growth, development, differentiation and apoptosis of cells, and acts as an essential component in various pathological processes [6]. In recent studies, CCL2 has also been found to promote angiogenesis [7]. PPARα is a nuclear receptor that affects lipid metabolism, the inflammatory response, and AS development by regulating the levels of transcriptions of various target genes through binding to PPARα ligands (such as fatty acids or BET inhibitors). It is also involved in maintaining blood sugar stability and enhancing tissue sensitivity to insulin [8–10]. Studies have shown that by directly acting on the arterial wall, PPARα can inhibit the migration of monocytes to vascular endothelial cells and their subsequent transformation into macrophages, and can block the proliferation and migration of VSMCs, prevent the formation of foam cells and reduce plaque instabilities [11,12]. However, the current research regarding the effect of PPARα inhibition on the pathogenesis and development of AS is limited to its roles in preventing the release of endothelial cell inflammatory factors and chemokines during the course of AS. The functional mechanism of PPARα in regulating the repair of endothelial cell injury remains unclear. Therefore, based on RNA-seq results and related data analysis, we conducted a detailed investigation of the regulatory interactions between CCL2 and PPARα and the role of such pathway in the repair of endothelial cell injury.

# 2. Material and methods

## 2.1. *In vitro* culture of HUVECs and the establishment of a lipid injury model

Human umbilical vein endothelial cells (HUVECs) were purchased from Qingyuanhao Biotechnology Co. Ltd. (Beijing, China). The cells were seeded into 6 cm culture dishes (Corning, NY, USA). When the cell growth density reached 100%, the cells were digested in 0.25% trypsin-ethylenediaminetetraacetic acid (Gibco, NY, USA) at 37°C for 3 min. After digestion, the cells were collected into a 15 ml centrifuge tube and centrifuged at $1500g$ for 5 min at 4°C. The supernatant was discarded, and 6 ml of endothelial cell growth basal medium (EBM-2) (Lonza, NY, USA) was added to obtain a single-cell suspension through gentle pipetting, followed by reseeding the cells into two 6 cm culture dishes and incubating at 37°C in a $CO_2$ incubator with 5% $CO_2$ and 95% humidity. The purchased HUVECs were passaged as described above and separated into two tubes. One tube was the untreated control cells (control group), while the other tube was treated with 200 µg ml$^{-1}$ of oxidized low-density lipoprotein (ox-LDL) and incubated for 24 h to induce the cell lipid peroxidative injury model (model group). Total RNA was extracted from all cells by adding 1 ml of TRIzol (Invitrogen, CA, USA) at the same time point.

## 2.2. Sample examinations and screening of differentially expressed genes

A standard RNA extraction protocol was followed to collect total RNA from the samples. The cells were collected and lysed in TRIzol before adding 200 µl of chloroform (China National Pharmaceutical Group, China) and allowed to stand for 10 min. The cells were then centrifuged at $12\,000g$ for 15 min. The upper aqueous phase was transferred to a new EP tube free of RNase and then mixed with 500 µl of isopropanol and allowed to stand for 10 min before centrifuging at $12\,000g$ for 10 min to precipitate the RNA. A white RNA pellet could be seen on the bottom of the tube. After discarding the supernatant, the RNA was washed with 75% ethanol and dried until the pellet became colourless and transparent. A total of 30 µl of diethyl pyrocarbonate $H_2O_2$ was added to resuspend the RNA, and the solution was stored at −80°C. The samples were divided into two groups: three samples of normal HUVECs were the control group and three samples of HUVECs with lipid injury were the cell injury group. Transcriptome sequencing was performed for all of the samples (Shanghai Fengxin Information Technology Co. Ltd.). Data comparisons and calculations of gene expression levels of the filtered sequences were conducted. The comparison of the transcriptome sequencing data was completed using TOPHAT v. 2.1.0 [13] and the genomic sequence information of *hg19* listed in the UCSC [14] database. Statistical analysis of the reads of known genes was completed using HTSeq [15] based on the *hg19* UCSC sequence. Finally, the degree distribution matrixes of the gene sequences in the two groups were obtained. Based on the expression matrixes of genes, differentially expressed genes (DEGs) between the two groups were analysed using the edgeR package in the R software [16]. Based on the significance threshold of

royalsocietypublishing.org/journal/rsob Open Biol. 9: 190141

$p < 0.05$ and on |logFC (fold change)| > 1, the significantly DEGs were identified.

## 2.3. Functional enrichment analysis of the genes and PPI network predictions

The functional enrichment analysis and pathway enrichment analysis of the DEGs were performed using the online tool DAVID [17]. The functional enrichment results in Biological Process and pathway enrichment results in KEGG were obtained [18]. The significance thresholds of enrichment used in the study were $p = 0.05$ for the modified Fisher exact test and a count >2. The PPIs between the DEGs were analysed using the online STRING database [19]. A required confidence (combined score) greater than 0.4 was used as the threshold for positive PPIs. The topological structure of the PPI network was analysed by Cytoscape [20] after obtaining the positive PPI pairs. From the biological network information attained, most biological networks were scale-free networks; thus, we applied connectivity degree analysis from network statistics to identify the important nodes in the PPI network that are involved in the PPIs (i.e. the protein hubs) [21].

## 2.4. Real-time quantitative PCR

The TB Green Premix Ex Taq kit (Takara, Tokyo, Japan) was used for all polymerase chain reaction (PCR). PCR was carried out using 1.0 µl of cDNA, corresponding to 500 ng of total RNA in a 20 µl final volume. After an initial 95°C for 2 min, 40 cycles of amplification were performed (95°C denaturing for 5 s, 60°C annealing and extension for 30 s). The reaction kinetics was represented by an amplification curve in which a region where the fluorescent increased exponentially was observed. A comparative threshold cycle ($\Delta C_T$) method was used to quantify target mRNAs and GAPDH was used as an internal reference gene. The primers used for reaction were as follows: NOX4 (sense 5′-AACCGAACCAGCTCTCAGAA-3′ and antisense 5′-AGCTTGGAATCTGGGCTCTT-3′), PPARA (sense 5′-CTGTCGGGATGTCACACAAC-3′ and antisense 5′-CGGG CTTTGACCTTGTTCAT-3′), CCL2 (sense 5′-GCTCAGCCA-GATGCAATCAA-3′ and antisense 5′-ACAGATCTCCTTGG CCACAA-3′), PDGFB (sense 5′-AAGACGTGGACTCCTCTT GG-3′ and antisense 5′-GTCACCATCTACAGCCACCT-3′), IL-8 (sense 5′-CTGGCAACCCTAGTCTGCTA-3′ and antisense 5′-AGTGCTTCCACATGTCCTCA-3′), VWF (sense 5′-GTGAG GCCTATGGCTTTGTG-3′ and antisense 5′-CGAGGTCAAGGT CCCTTCTT-3′), CD36 (sense 5′-ACTCAGTGTTGGTGTGGT GA-3′ and antisense 5′-ATGCAGGGCCTAGGATTTGT-3′) and GAPDH (sense 5′-ACCCAGAAGACTGTGGATGG-3′ and antisense 5′-TCAGCTCAGGGATGACCTTG-3′).

## 2.5. Western-blot analysis

HUVECs were collected using RIPA buffer (Beyotime, Shanghai, China) and were sonicated at 4°C for 30 s after it was incubated on ice for 30 min. The supernatant was collected after centrifuging at 12 000$g$ for 10 min at 4°C for BCA protein quantification (Beyotime, Shanghai, China). Western blot analysis was performed as described previously. Primary polyclonal anti-CCL2 (1 : 1000), anti-NOX4 (1 : 2000), anti-PPARα (1 : 2000), anti-PDGFβ (1 : 1000), anti-VWF (1 : 2000) and anti-β-actin (1 : 5000) antibodies were all purchased from Abcam (Abcam, MA, USA). The primary antibody was added to the PVDF membrane for incubation overnight at 4°C; after the membrane was washed three times for 15 min with TBST at room temperature (RT), it was incubated with a secondary horseradish peroxidase-conjugated goat anti-rabbit or goat anti-mouse immunoglobulin (Ig)G antibody (1 : 2000; CST, MA, USA) for 2 h at RT. The western blot bands were visualized using the enhanced chemiluminescence kit (Beyotime, Shanghai, China) and quantified via densitometric analysis using IMAGE-PRO PLUS v. 6.0 software (Media Cybernetics, USA).

## 2.6. Luciferase reporter assay

All EA.hy926 cells (Cell Bank, Shanghai Institute for Biological Science, Shanghai, China) were seeded into six-well cell culture plates at a density of $1 \times 10^5$ cells well$^{-1}$. The Lipofectamine 3000 Reagent was used to transfect the cells in the respective groups with 2500 ng of PDS131_psiCHECK-2 blank vector (6273 bp), PDS131_psiCHECK-2-CCL2 wild-type (wt, gene synthesis of the CCL2 gene promoter region 1900 bp and subcloning into the reporter vector) and PDS131_psiCHECK-2-CCL2 mutant (mut, deletion mutation of 'TTGCCTCAGTG' in the promoter region of wild-type vector CCL2 gene) (Novobio Biotechnology Co. Ltd., Shanghai, China). At 72 h after transfection, the dual-luciferase reporter assay system (Beyotime Biotechnology Co. Ltd., Zhejiang, China) was used to detect the luciferase activity in each group. We first use the microplate reader to detect the corresponding value of the fluorescence emitted by the firefly luciferase-inducing substrate (FL), then use the microplate reader to detect the corresponding value of the fluorescence released by the renilla luciferase-inducing substrate (RL), the ratio of the value of FL to RL measured for each sample as the relative luciferase activity of the reporter gene.

## 2.7. Immunofluorescence staining

$Apoe^{-/-}$ mice (2-month-old, male) on a normal diet (control group), $Apoe^{-/-}$ mice on a high-fat diet (model group), pemafibrate-treated (MedChemExpress, USA) $Apoe^{-/-}$ mice on a high-fat diet (PPARα agonist group) and GW6471-treated (APExBIO, USA) $Apoe^{-/-}$ mice on a high-fat diet (PPARα antagonist group) were used to detect whether PPARα affects the expression of CCL2 in endothelial cells in vivo. The total treatment time was three months for each group; for the drug groups, mice simultaneously were given the drug while on the high-fat diet. All $Apoe^{-/-}$ mice were kept in the SPF grade animal facility at the animal centre of the Shanghai University of Traditional Chinese Medicine. The RT was maintained at 24°C with a relative humidity of 50–60%, and a 12 L : 12 D cycle was used. All tissue sections were frozen, and the left ventricular outflow tract was sliced. Briefly, after incubation with a blocking buffer (Beyotime, Shanghai, China) for 30 min at RT, the sections were incubated with the primary antibodies against CD31 (1 : 300, Boster, Wuhan, China) and CCL2 (1 : 300, Boster, Wuhan, China) overnight at 4°C. On the following day, the sections were incubated with goat anti-rabbit IgG (H + L) (Alexa Fluor 546) and rabbit anti-mouse IgG (H + L) (Alexa Fluor 488) (Invitrogen, MA, USA). Sections were washed, mounted and examined using a fluorescence microscope (CX43 Biological Microscope, Olympus, Japan).

## 2.8. Construction of stable HUVEC lines with transfected PPARα and CCL2

Lentivirus with overexpressed and knockdown PPARα or CCL2 was purchased from Shanghai Gene Chemical Technology Co., Ltd., and the virus titre was $1 \times 10^8$ TU ml$^{-1}$. Cell cultures were prepared by seeding HUVECs at $5 \times 10^6$ cells ml$^{-1}$ into 10 cm culture dishes. When the cell density reached greater than 90%, 400 μl of virus infection-enhancing solution HitransG A and 100 μl of lentivirus with a titre of $1 \times 10^8$ TU ml$^{-1}$ were added, and the cells were cultured at 37°C. The media was refreshed every 24 h. At 72 h after infection, the cells were inspected for the rate of infection (the number of green fluorescent protein-expressing cells). Once the rate of infection reached greater than 90%, the cells were used in subsequent experimental procedures. Cell groups with overexpressed PPARα or CCL2 were labelled as PPARα-ov or CCL2-ov, respectively; cells with interfered PPARα or CCL2 were labelled as PPARα RNAi or CCL2 RNAi, respectively.

## 2.9. Terminal deoxyribonucleotidyl transferase-mediated dUTP-digoxigenin nick end labelling assay for apoptosis

HUVECs seeded in 24-well plates were fixed with 4% paraformaldehyde for 15 min after removing the culture medium. After fixation, the cells were washed three times with phosphate-buffered saline (PBS), then PBS containing 0.3% Triton X-100 was added, and the cells were incubated for 5 min at RT to remove the PBS. Next, 50 μl of terminal deoxyribonucleotidyl transferase-mediated dUTP-digoxigenin nick end labelling (TUNEL) assay solution (Beyotime, Shanghai, China) was added, and the cells were incubated at 37°C for 60 min in the dark and washed three times with PBS. The cells were sealed with an anti-fluorescence quenching liquid and observed under a fluorescence microscope.

## 2.10. Detection of cell proliferation using Ki67 staining

HUVECs were seeded into a 24-well plate at a seeding density of approximately $1 \times 10^4$ cells well$^{-1}$. After the previous experiments and fixation, ki67 rabbit polyclonal antibody (1 : 1000, Beyotime, Shanghai, China) was added to each well at 4°C overnight and washed three times with PBS, then goat anti-rabbit Alexa Fluor 546 (Invitrogen, MA, USA) was added to each well for 60 min at 37°C in the dark and washed three times with PBS. The cells were sealed with an anti-fluorescence quenching liquid and observed under a fluorescence microscope.

## 2.11. Transwell assays

*In vitro* Transwell assays were used to detect the migration ability of HUVECs in the different treatment groups. In the Transwell assay, we measured the migration ability of HUVECs using a 5.6 mm polycarbonate membrane (with 8 μm pores) on a Transwell chamber (Corning, NY, USA). A total of 500 μl of Dulbecco's modified Eagle's medium containing 10% FBS was added to the lower chamber of a 24-well plate, and 200 μl of $1 \times 10^5$ HUVEC solutions diluted in EBM-2 was placed in the upper chamber and cultured in

**Table 1.** Identification of differentially expressed genes (DEGs).

| | gene counts |
| --- | --- |
| upregulated | 258 |
| downregulated | 53 |
| total | 311 |

an incubator for 24 h. Next, unmigrated cells on the upper surface of the chamber membrane were removed with a cotton swab, the cells that had migrated to the lower surface of the chamber were fixed and stained with crystal violet, and the number of migrated cells was counted under a microscope.

## 2.12. Angiogenesis assay

The angiogenesis assay is a routine test to examine the angiogenic ability of HUVECs. Briefly, HUVECs were seeded in Matrigel-coated μ-Slide angiogenesis plates (Ibidi, Martinsried, Germany) at $1 \times 10^4$ cells well$^{-1}$, and 20 ng ml$^{-1}$ VEGF165 was added to the cell culture medium. The formation of capillary-like structures was inspected under a phase contrast microscope 18 h later. The number of branches in the generated blood vessels was calculated and compared using ImageJ software.

## 2.13. Statistical analysis

Statistical analysis was performed using SPSS v. 18.0 software (SPSS Inc., Chicago, USA). All data are presented as mean ± s.d. One-way ANOVA and Tukey HSD *post hoc t*-tests were used to determine the level of statistical significance. $p < 0.05$ was considered a statistically significant difference.

# 3. Results

In order to further study the regulational mechanism of endothelial cell injury and repair, we established a vascular endothelial cell injury model *in vitro* featuring endothelial cell lipid peroxidative injury, and used RNA-seq and relevant data analysis to identify genes involved in the repair of endothelial cell injury. Cultured HUVECs treated with 200 μg ml$^{-1}$ of ox-LDL were used as a cell model, transcriptome sequencing was performed and 311 DEGs were analysed after collecting RNA samples from model and control HUVECs.

## 3.1. Identification of differentially expressed genes and PPI network predictions

The results of the DEGs obtained by the aforementioned methods and the significance thresholds are shown in table 1 and figure 1, and electronic supplementary material, table S1. Compared with the control group, there were 258 upregulated genes and 53 downregulated genes in the cell injury group. We used STRING and the corresponding parameter settings to analyse the PPI relationships of the genes and acquired 181 relationship pairs (figure 2). Next, we performed network statistical analysis on the PPI relationships of the DEGs. Table 2 lists the top 20 most highly connected protein nodes within the PPI network; italic type represents the downregulated genes, bold

royalsocietypublishing.org/journal/rsob   Open Biol. **9**: 190141

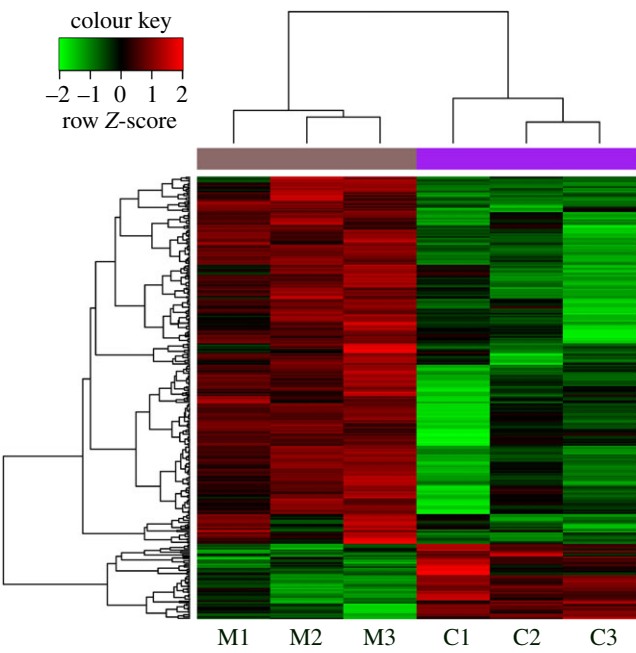

**Figure 1.** Results of hierarchical clustering analysis (heat-map). Red and green in the colour bar represent high and low expression levels, respectively. Red and green in the y-axis represent upregulated and downregulated DEGs, respectively. DEGs were analysed between model group (M1, M2 and M3) and control group (C1, C2 and C3).

type denotes the upregulated genes and degree represents the degree of connectivity of the genes in the network. The interactions between PPI-predicted proteins include direct (physical) and indirect (functional) associations.

## 3.2. Functional enrichment analysis of the genes and verification of the expression levels of the highly connected genes

GO functional enrichment analysis and KEGG pathway enrichment analysis were performed on the DEGs (up- or downregulated). The results of the analysis are shown in tables 3 and 4, respectively showing the enrichment results of the upregulated and downregulated genes. No significant enrichment of the downregulated genes was detected in the KEGG pathway analysis.

Most of the PPI network nodes with high connectivity were upregulated genes, and many genes involved in endothelial cell lipid peroxidative injury (GO: 0009611-response to wounding) and its repair (GO: 0042060-wound healing) (table 5) were also upregulated genes. Therefore, we screened the upregulated genes (NOX4, PPARA, CCL2, PDGFB, IL8, VWF, CD36) that were associated with cell damage repair and were also highly connected nodes in the PPI network. The real-time PCR results demonstrated that messenger RNA (mRNA) expression levels of the genes associated with HUVEC lipid peroxidative injury were significantly higher than those of cells in the control group, and the difference was significant (figure 3a).

## 3.3. Selective PPI network predictions and protein functional assays

According to the upregulated genes with a high degree of node connectivity in the PPI network and the results of

gene functional enrichment analysis related to injury repair, we examined potential PPIs between these genes using the STRING database with corresponding parameter settings and generated a new PPI network prediction (figure 3b) after removing the genes without PPIs or with known interactions. We found that among the relevant genes verified by real-time PCR, including NOX4, PPARA, CCL2, PDGFB, IL8, VWF and CD36, CCL2 was predicted to have interactions with many other genes (NOX4, PPARA, PDGFB and VWF). Thus, we examined the protein expression levels of CCL2, NOX4, PPARα, PDGFβ and VWF (figure 3c). The results confirmed that the expression levels of the above proteins were all upregulated after endothelial cell injury, but the increases in CCL2 and PPARα proteins were the most remarkable. Therefore, we speculate that CCL2 and PPARα play critical roles in the injury repair caused by lipid peroxidation in HUVECs.

## 3.4. Roles of PPARα on regulating CCL2 expression

We first constructed CCL2- and PPARα-overexpressed or knock-down HUVECs to verify the regulatory relationship between CCL2 and PPARα. It was discovered that CCL2-ov or CCL2 RNAi did not change the expression levels of PPARα in HUVECs (figure 4a); however, the CCL2 expression was significantly elevated after introducing PPARα-ov into the cells, and the expression of CCL2 was significantly decreased after PPARα RNAi treatment (figure 4b). Consequently, we need to detect whether CCL2 is regulated by PPARα at transcription or post-transcription level. The results of real-time PCR indicated that the mRNA level of CCL2 was also significantly upregulated after introducing PPARα-ov into the cells while the mRNA level of CCL2 was significantly downregulated after PPARα RNAi treatment. After clarifying that CCL2 is regulated by PPARα at the transcriptional level, we performed verification of PPARα consensus binding site in the promoter region of CCL2 gene in order to distinguish whether CCL2 is regulated by PPARα directly or indirectly. We constructed the promoter region of CCL2 into the PDS131_psiCHECK-2 plasmid containing the dual-luciferase reporter gene. We also constructed the promoter region of CCL2 lacking the TTGCCTCAGTG fragment into the PDS131_psiCHECK-2 plasmid, and we transfected blank plasmid, wild-type plasmid and mutant plasmid into EA.hy926 cell line. EA.hy926 is a human umbilical vein cell fusion cell line, and has the biological characteristics of HUVECs and expressed endogenous PPARα. The luciferase reporter gene assay showed that the deletion of the CCL2 promoter fragment resulted in the inability of combination of PPARα and CCL2; however, the strong fluorescence was detected in wild-type plasmid transfected cells, indicating that PPARα can directly bind to the CCL2 promoter and promote CCL2 expression (figure 4c).

Accordingly, $Apoe^{-/-}$ mice with normal diet as a control group and $Apoe^{-/-}$ mice with high-fat diet as a model group were used to examine PPARα and CCL2 expression of the left ventricular outflow tract tissues (PPARα agonist and antagonist only treated $Apoe^{-/-}$ mice on a high-fat diet). We further validated whether PPARα affects the expression of CCL2 in endothelial cells in vivo, compared with the $Apoe^{-/-}$ model group mice, PPARα agonist significantly promoted the expression of PPARα and CCL2, while PPARα antagonist significantly inhibited the expression of PPARα and CCL2, and there was no significant difference

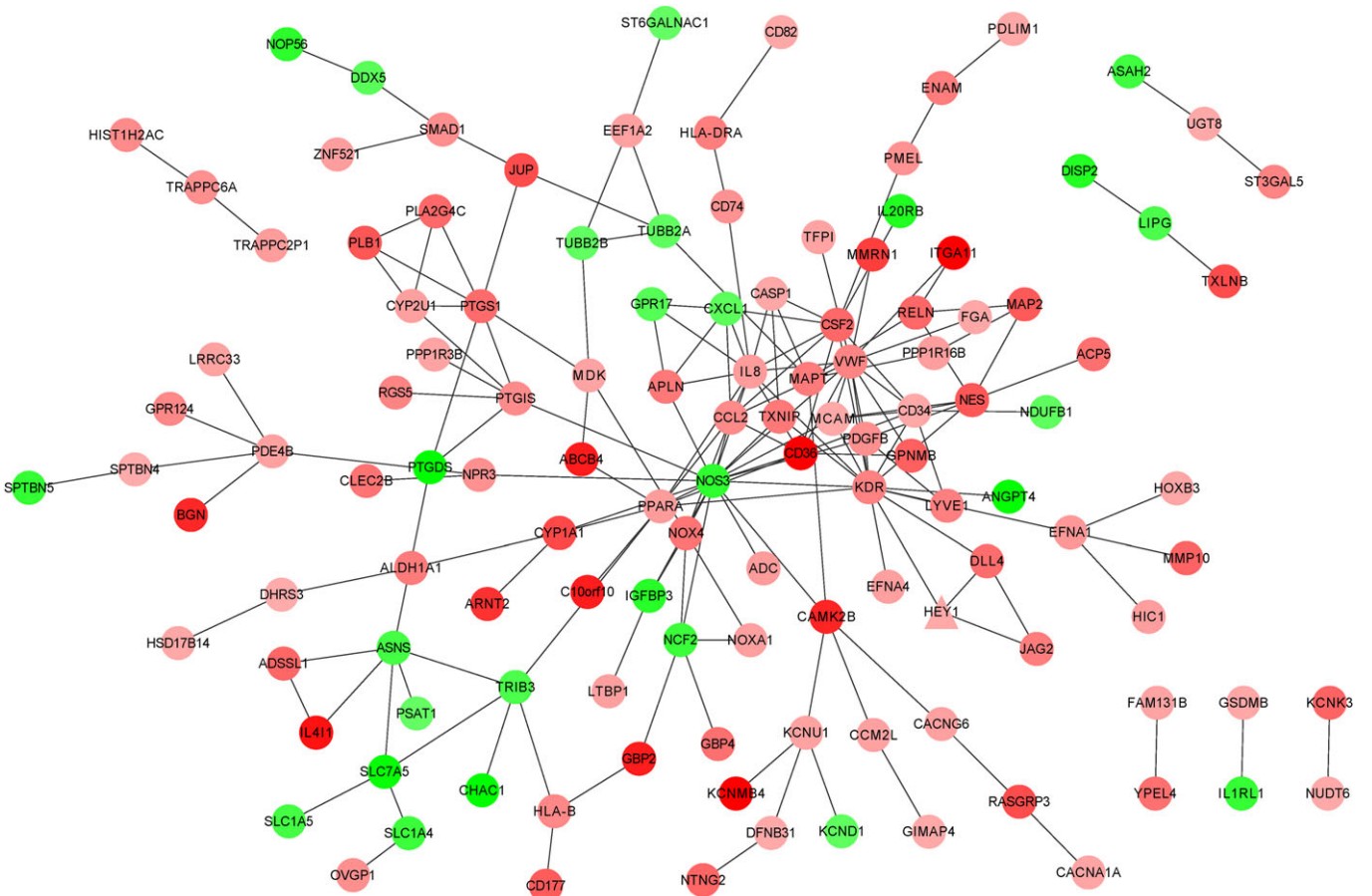

**Figure 2.** PPI network of DEGs. Red nodes represent DEGs upregulated in the model group, while green nodes represent DEGs downregulated in the model group. The colour shade of a red node is positively correlated with the degree of this node; the darker the colour is, the higher the expression of upregulational DEGs is.

**Table 2.** Top 20 most highly connected protein nodes within the protein–protein interaction (PPI) network. Bold and italics in the table represent upregulated and downregulated DEGs, respectively. Degree represents the degree of connectivity of the genes in the network.

| DEG | degree | DEG | degree | DEG | degree | DEG | degree |
|---|---|---|---|---|---|---|---|
| *NOS3* | *18* | **CSF2** | **9** | **PTGS1** | **7** | **MAPT** | **5** |
| **KDR** | **16** | **CD34** | **8** | **NOX4** | **6** | **MCAM** | **5** |
| **VWF** | **13** | **CCL2** | **8** | **PTGIS** | **6** | *TRIB3* | *5* |
| **IL8** | **12** | **CD36** | **8** | *ASNS* | *6* | *NCF2* | *5* |
| **PPARA** | **9** | **NES** | **7** | **PDGFB** | **5** | **PDE4B** | **5** |

in PPARα and CCL2 expression between the $Apoe^{-/-}$ control group and the $Apoe^{-/-}$ model group; in endothelial cells of the left ventricular outflow tract tissues, we also found that PPARα agonists significantly promote the expression of CCL2 in endothelial cells and we only detected a very weak expression of CCL2 in the PPARα antagonist group (figure 4d). Therefore, we speculate that PPARα may participate in the repair of cell injury induced by lipid peroxidation through regulating CCL2 expression.

## 3.5. PPARα participates in the repair of cell injury induced by lipid peroxidation through regulating CCL2 expression

Next, we examined the effects of PPARα upregulation and downregulation on the function of HUVECs. Interestingly, in PPARα-OV HUVECs, the number of apoptotic cells was reduced, while cell proliferation, migration and angiogenic abilities were all enhanced. As expected, inhibiting PPARα expression produced the opposite effects on the above cell functions in HUVECs (figure 5). To further investigate whether the above functions of PPARα were achieved by regulating CCL2, we examined the functions of HUVECs after adding PPARα agonist pemafibrate to the stable CCL2 knockdown HUVECs. First, we detected CCL2 expression after adding ox-LDL or pemafibrate to the stable CCL2 knockdown HUVECs. ox-LDL inhibited CCL2 expression contrast to the control group, while pemafibrate enhanced CCL2 expression contrast to the control group; in the stable CCL2 knockdown HUVECs, we detected a very weak expression of CCL2 with or without ox-LDL and pemafibrate (figure 6a). The results about the function of HUVECs showed that when CCL2 expression was inhibited, the ability of PPARα to promote HUVECs proliferation and

royalsocietypublishing.org/journal/rsob   Open Biol. 9: 190141

**Table 3.** Pathway enrichment analysis of upregulated DEGs.

| category | description | count | p-value |
|---|---|---|---|
| BP | GO:0009611~response to wounding | 23 | $3.52 \times 10^{-7}$ |
| BP | GO:0042060~wound healing | 12 | $1.86 \times 10^{-5}$ |
| BP | GO:0050817~coagulation | 8 | $2.21 \times 10^{-4}$ |
| BP | GO:0007596~blood coagulation | 8 | $2.21 \times 10^{-4}$ |
| BP | GO:0007599~haemostasis | 8 | $3.14 \times 10^{-4}$ |
| BP | GO:0007155~cell adhesion | 20 | $7.13 \times 10^{-4}$ |
| BP | GO:0022610~biological adhesion | 20 | $7.25 \times 10^{-4}$ |
| BP | GO:0001666~response to hypoxia | 8 | $1.14 \times 10^{-3}$ |
| BP | GO:0070482~response to oxygen levels | 8 | $1.53 \times 10^{-3}$ |
| BP | GO:0050878~regulation of body fluid levels | 8 | $1.53 \times 10^{-3}$ |
| CC | GO:0031012~extracellular matrix | 16 | $6.52 \times 10^{-5}$ |
| CC | GO:0044459~plasma membrane part | 50 | $1.57 \times 10^{-4}$ |
| CC | GO:0005886~plasma membrane | 72 | $6.37 \times 10^{-4}$ |
| CC | GO:0009986~cell surface | 14 | $8.56 \times 10^{-4}$ |
| CC | GO:0005576~extracellular region | 44 | 0.001035 |
| CC | GO:0044421~extracellular region part | 26 | 0.001068 |
| CC | GO:0031226~intrinsic to plasma membrane | 30 | 0.001624 |
| CC | GO:0005887~integral to plasma membrane | 28 | 0.004688 |
| CC | GO:0043025~cell soma | 8 | 0.007644 |
| CC | GO:0009897~external side of plasma membrane | 8 | 0.008177 |
| MF | GO:0005509~calcium ion binding | 21 | 0.004862 |
| MF | GO:0005518~collagen binding | 4 | 0.008315 |
| MF | GO:0005200~structural constituent of cytoskeleton | 5 | 0.010781 |
| MF | GO:0005539~glycosaminoglycan binding | 6 | 0.023757 |
| MF | GO:0005198~structural molecule activity | 14 | 0.033515 |
| MF | GO:0030247~polysaccharide binding | 6 | 0.033906 |
| MF | GO:0001871~pattern binding | 6 | 0.033906 |
| MF | GO:0019838~growth factor binding | 5 | 0.034069 |
| MF | GO:0009055~electron carrier activity | 7 | 0.044396 |
| KEGG | hsa04512:ECM-receptor interaction | 5 | $2.30 \times 10^{-2}$ |

migration and angiogenesis was significantly reduced, while apoptotic cells increased significantly in these HUVECs (figure 6b–e). Based on the above data, we conclude that PPARα promotes the repair of endothelial cell injury through upregulating the expression of CCL2 in HUVECs.

## 4. Discussion

Vascular endothelial cells are semipermeable membrane barriers between the blood and the sub-endothelial tissues, and have perceptual and secretory functions. Physical damage to these cells increases the permeability of the endothelium to lipoproteins and other plasma components [22]. As a result, nitric oxide secretion is reduced, secretions of intercellular adhesion molecules and vascular cell adhesion molecules are increased, and nuclear transcription factor expression is elevated, rendering the cells to a pro-inflammatory and pro-apoptotic state [23,24]. It has been shown in animal experiments that branches or branching points of blood vessels are clearly susceptible to haemodynamic shears and tensile changes, which leads to chronic damage to the endothelial cells. To maintain their morphological and functional integrity to withstand such damage, endothelial cells increase their turnover rates, which leads to the accumulation of aging endothelial cells and an increased susceptibility to developing AS plaques [25]. Platelet-derived growth factor (PDGF) secreted by endothelial cells can promote smooth muscle cell chemotaxis and proliferation; inflammatory factors released by these cells can recruit a large number of monocytes and polymorphonuclear neutrophils. ICAM-1 and VCAM-1 secreted by endothelial cells can promote the adhesion of monocytes to endothelial cells, increasing the chance of monocyte exudation. As monocytes play a direct role in the development of AS, it is reasonable to consider endothelial cell injury as a key component in the development of AS [26,27].

Based on the above findings, we generated an in vitro ox-LDL injury model of cultured vascular endothelial cells. We predicted PPIs between DEGs that are significantly upregulated in

**Table 4.** Pathway enrichment analysis of downregulated DEGs.

| category | description | count | *p*-value |
|---|---|---|---|
| BP | GO:0015804~neutral amino acid transport | 3 | $1.11 \times 10^{-3}$ |
| BP | GO:0008285~negative regulation of cell proliferation | 5 | $9.94 \times 10^{-3}$ |
| BP | GO:0006865~amino acid transport | 3 | $1.91 \times 10^{-2}$ |
| BP | GO:0015837~amine transport | 3 | $3.16 \times 10^{-2}$ |
| BP | GO:0048662~negative regulation of smooth muscle cell proliferation | 2 | $3.26 \times 10^{-2}$ |
| BP | GO:0006835~dicarboxylic acid transport | 2 | $3.49 \times 10^{-2}$ |
| BP | GO:0043200~response to amino acid stimulus | 2 | $3.95 \times 10^{-2}$ |
| BP | GO:0046942~carboxylic acid transport | 3 | $4.70 \times 10^{-2}$ |
| BP | GO:0015849~organic acid transport | 3 | $4.76 \times 10^{-2}$ |
| CC | GO:0005856~cytoskeleton | 10 | $1.02 \times 10^{-2}$ |
| CC | GO:0044430~cytoskeletal part | 8 | $1.33 \times 10^{-2}$ |
| CC | GO:0043232~intracellular non-membrane-bounded organelle | 13 | $3.81 \times 10^{-2}$ |
| CC | GO:0043228~non-membrane-bounded organelle | 13 | $3.81 \times 10^{-2}$ |
| MF | GO:0015175~neutral amino acid transmembrane transporter activity | 3 | $1.13 \times 10^{-3}$ |
| MF | GO:0015171~amino acid transmembrane transporter activity | 3 | $8.67 \times 10^{-3}$ |
| MF | GO:0005275~amine transmembrane transporter activity | 3 | $1.34 \times 10^{-2}$ |
| MF | GO:0048037~cofactor binding | 4 | $2.11 \times 10^{-2}$ |
| MF | GO:0017153~sodium:dicarboxylate symporter activity | 2 | $2.36 \times 10^{-2}$ |
| MF | GO:0042605~peptide antigen binding | 2 | $3.52 \times 10^{-2}$ |
| MF | GO:0005310~dicarboxylic acid transmembrane transporter activity | 2 | $3.75 \times 10^{-2}$ |
| MF | GO:0008047~enzyme activator activity | 4 | $4.49 \times 10^{-2}$ |

**Table 5.** Upregulated DEGs associated with cell damage and repair in biological processes of gene ontology (GO) functional enrichment analysis.

| biological process | |
|---|---|
| **GO:0009611~response to wounding** | **GO:0042060~wound healing** |
| IRAK2, NOX4, F11R, PPARA, CCL2, CYP1A1, PDGFB, IL8, SCUBE1, SMAD1, MMRN1, MDK, CHST1, APOL3, VWF, LYVE1, DYSF, CD36, PLSCR4, FGA, TFPI, PLA2G4C, TFPI2 | VWF, PPARA, CD36, DYSF, PLSCR4, FGA, PDGFB, SCUBE1, TFPI, SMAD1, MMRN1, TFPI2 |

response to endothelial cell injury repair using RNA-seq and related analyses. The results found that CCL2 is a central protein that interacts with multiple other proteins. After examining the expression levels of multiple related proteins, we focused on CCL2 and PPARα, as the two showed the most substantial increases in protein expression levels. The expression of CCL2, a chemokine, is elevated in AS lesions [28]. Knocking out CCL2 in $LDLR^{-/-}$ or $ApoE^{-/-}$ mice delays the development of AS [29,30]. It is believed that VSMCs and monocytes in AS lesions secrete a large amount of CCL2 to promote the migration of monocytes to the lesion, where the monocytes fuse with vascular endothelial cells and transform into foam cells [31]. In addition, CCL2 has also been found to be critically involved in promoting angiogenesis. Thus, CCL2 may play a dual regulatory role in the development of AS. In this study, we found that oxidative damage to vascular endothelial cells enhanced their ability to secrete CCL2. The involvement of CCL2 in the repair of vascular endothelial cell damage was also revealed in the GO functional enrichment analysis. Therefore, we speculate that in the early stage of cell injury, CCL2 is a primary participant in the injury repair mechanism of endothelial cells, while in the formation of AS, CCL2 is mostly involved in the recruitment of monocytes to engulf oxycholesterols. The essence of these two functions of CCL2 is to prevent the development of AS, but when CCL2 fails to repair the damaged endothelial cells or the monocytes recruited by CCL2 fail to engulf large amounts of oxycholesterols, the development of AS becomes inevitable.

To further understand the regulatory relationship between CCL2 and PPARα, we generated stably transfected endothelial cell lines with overexpressed or knockdown CCL2 and PPARα. It was found that the expression of CCL2 did not affect PPARα expression, whereas the PPARα expression was positively correlated with CCL2 expression, indicating that PPARα regulates CCL2 secretion. PPARα is a nuclear receptor that regulates the transcription level of multiple target genes by binding to the ligands. Studies have shown that PPARα can inhibit the migration of monocytes towards vascular endothelial cells and their subsequent transformation into macrophages, block the proliferation and migration of VSMCs, prevent foam cell formation and reduce the instability of plaques by directly acting on the arterial wall. The functions of PPARα in AS are similar to those of CCL2, but an in-depth

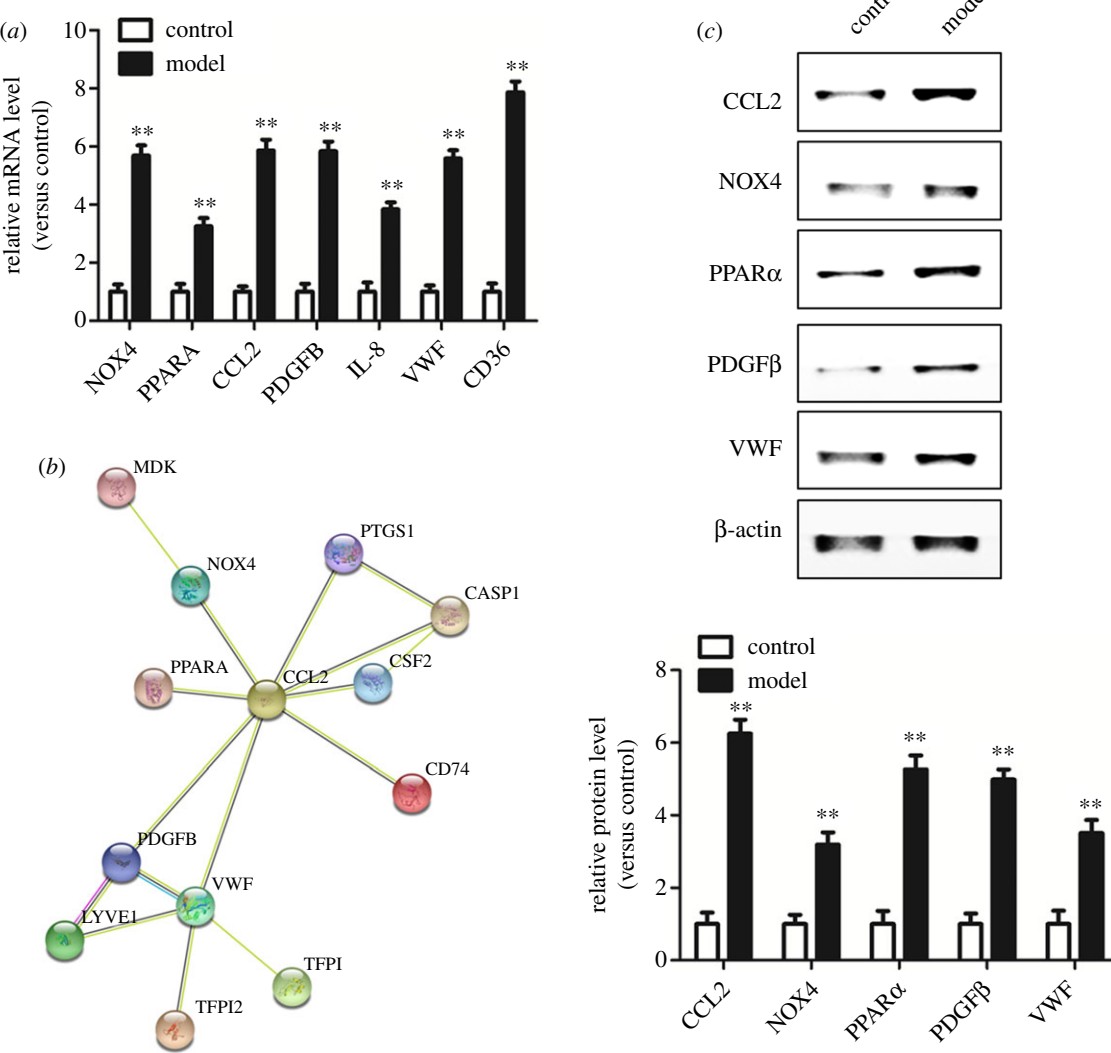

**Figure 3.** Real-time PCR and western blot verify the upregulated genes associated with damage repair and high connectivity in PPI network nodes. (*a*) We screened the upregulated genes (NOX4, PPARA, CCL2, PDGFB, IL8, VWF, CD36) that were associated with cell damage repair and were also highly connected nodes in the PPI network. The real-time PCR results demonstrated that expression levels of DEGs associated with HUVEC lipid peroxidative injury were significantly higher than those of cells in the control group. (*b*) We generated a new PPI network prediction using the STRING database according to tables 2 and 5, we found that CCL2 was predicted to have interactions with other genes (NOX4, PPARα, PDGFβ and VWF). (*c*) Western-blot examined the protein expression levels of CCL2, NOX4, PPARα, PDGFβ and VWF, and the increases in CCL2 and PPARα proteins were the most remarkable. $**p < 0.01$ (versus control).

examination of the role of PPARα in the repair of vascular endothelial cell injury is still lacking. We knocked out the expression of PPARα in HUVECs and found that apoptosis of the endothelial cells increased, while cell proliferation decreased, along with a reduced migration and angiogenesis ability. After adding PPARα agonist into the si-CCL2 HUVECs, the same phenomenon as discovered in PPARα knockout HUVECs was observed. Thus, we conclude that PPARα plays a critical role in inhibiting apoptosis and promoting cell proliferation, migration and angiogenesis, and these functions are achieved through its regulation of CCL2.

In this study, RNA-seq assays and relevant analyses of endothelial cell lipid peroxidative injury revealed that PPARα and CCL2, two significantly upregulated genes after injury, may be involved in the repair of injured endothelial cells. Previous studies on PPARα and CCL2 have predominantly focused on the chemotaxis to monocytes and on the formation of foam cells. In this study, we used endothelial cells as independent target cells to examine the regulatory relationship between PPARα and CCL2 and their functions in the repair of injured endothelial cells. We found that

PPARα can promote the repair of endothelial cells by upregulating the expression of CCL2. Therefore, we believe that PPARα and CCL2 play critical roles in the repair of injured endothelial cells during the early stage of AS development.

# 5. Conclusion

In our research, RNA sequencing technology was used for a transcriptome study of endothelial cell lipid peroxidative injury, followed by protein interactions analysis on the DEGs, protein functional research and related mechanisms. First, we found 311 DEGs. Using the STRING database and GO functional enrichment analysis, we predicted that CCL2 interacts with NOX4, PPARα, PDGFβ and VWF individually. Consequently, according to the results of protein expression analysis, we investigated the relationship between CCL2 and PPARα, and their roles in the repair of endothelial cell injury. Molecular mechanism study further confirmed that PPARα participates in the repair of endothelial cell lipid peroxidative injury through regulating the expression of CCL2.

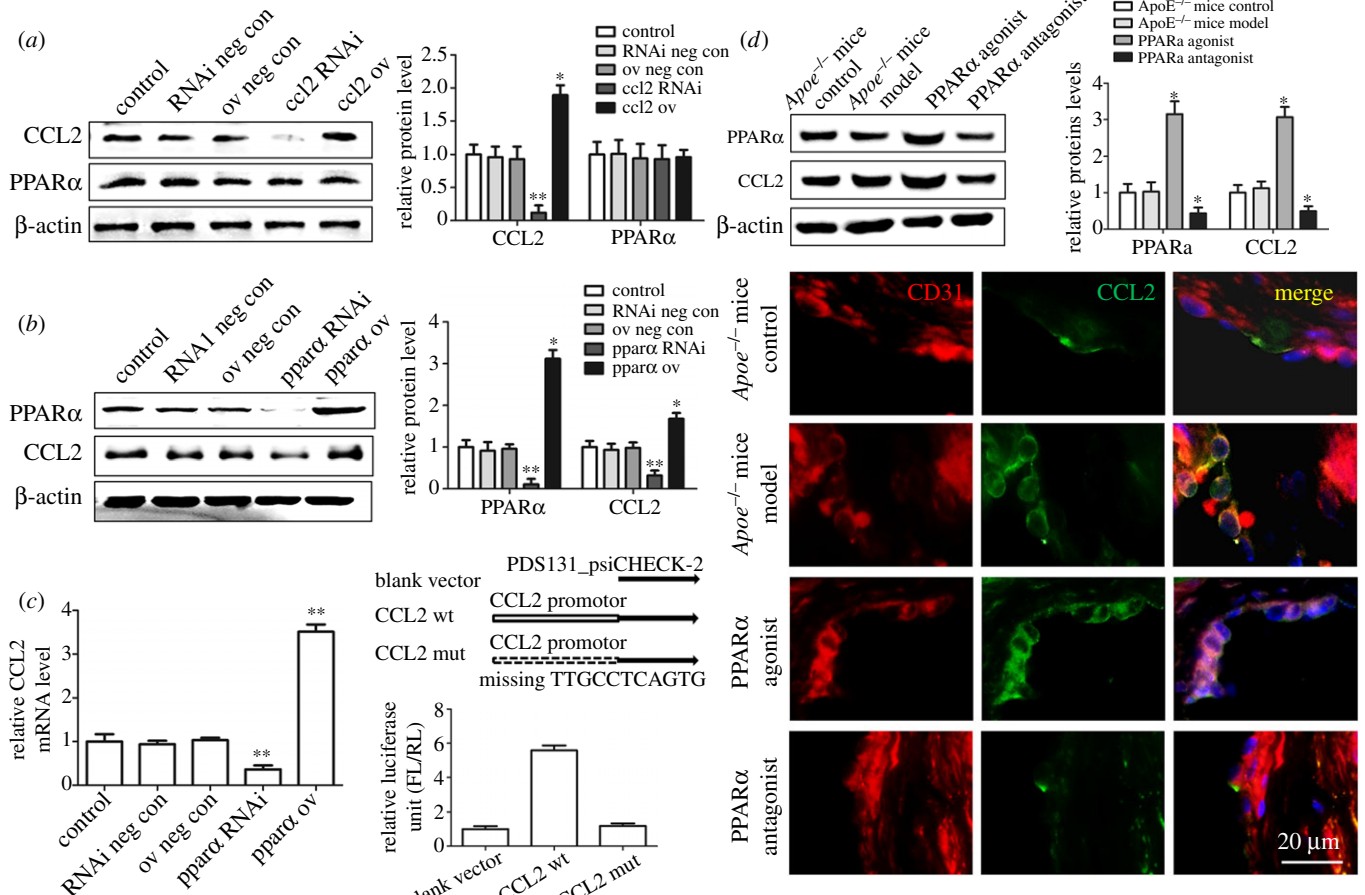

**Figure 4.** Western blot and immuno-histochemical staining verify the regulatory relationship between CCL2 and PPARα. (a) CCL2-ov (CCL2 overexpression) or CCL2 RNAi (CCL2 knockdown) did not change the expression levels of PPARα in HUVECs. (b) CCL2 expression was significantly elevated after introducing PPARα ov (PPARα overexpression) into the cells, and the expression of CCL2 was significantly decreased after PPARα RNAi (PPARα knockdown) treatment. (c) The mRNA level of CCL2 was also significantly upregulated after PPARα ov treatment while the mRNA level of CCL2 was significantly downregulated after PPARα RNAi treatment; the luciferase reporter gene assay showed that deletion of the CCL2 promoter fragment resulted in the inability of the combination of PPARα and CCL2. (d) PPARα agonist significantly promoted the expression of PPARα and CCL2, while PPARα antagonist significantly inhibited the expression of PPARα and CCL2, at the same time, PPARα agonists promoted the expression of CCL2 in endothelial cells of the left ventricular outflow tract tissues in ApoE$^{-/-}$ mice. *$p < 0.05$, **$p < 0.01$ (versus control). Scale bar, 20 μm.

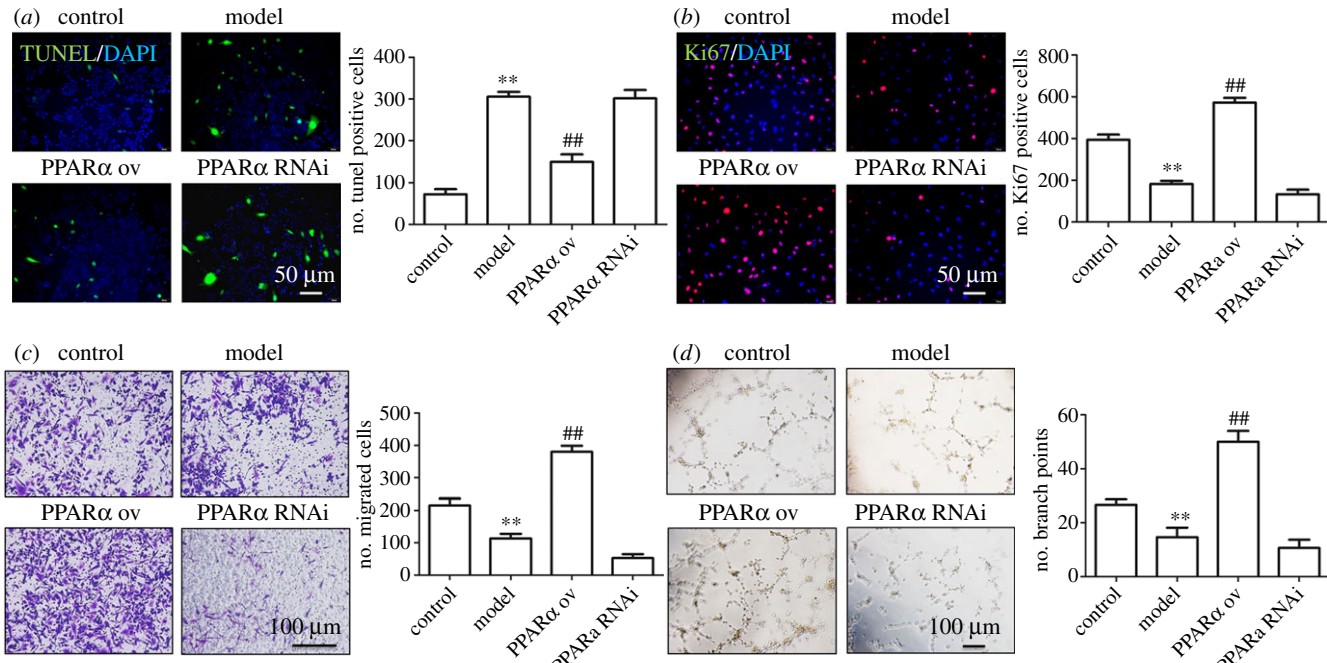

**Figure 5.** PPARα participates in the repair of cell injury induced by lipid peroxidation: (a) PPARα overexpression inhibits HUVEC apoptosis, (b) PPARα overexpression promotes HUVEC proliferation, (c) PPARα overexpression promotes HUVEC migration and (d) PPARα overexpression promotes angiogenesis. **$p < 0.01$ (versus control); ##$p < 0.01$ (versus model). Scale bar, 50 μm or 100 μm.

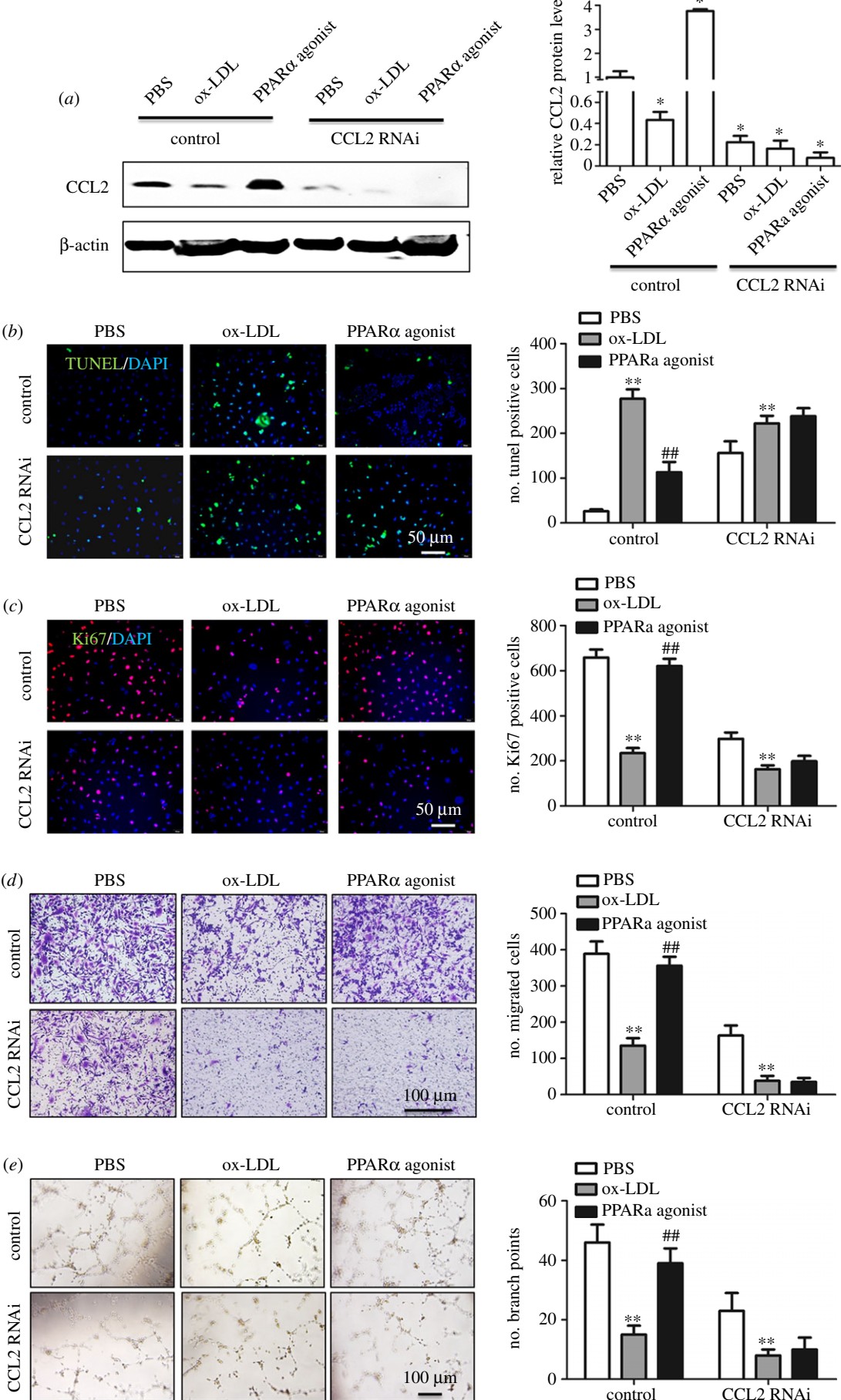

**Figure 6.** PPARα participates in the repair of cell injury induced by lipid peroxidation through regulating CCL2 expression: (*a*) western-blot assay examined the CCL2 expression after adding ox-LDL or pemafibrate to the stable CCL2 knockdown HUVECs, (*b*) PPARα did not inhibit HUVEC apoptosis when CCL2 expression was inhibited, (*c*) PPARα did not promote HUVEC proliferation when CCL2 expression was inhibited, (*d*) PPARα did not promote HUVEC migration when CCL2 expression was inhibited and (*e*) PPARα did not promote angiogenesis when CCL2 expression was inhibited. *$p < 0.05$, **$p < 0.01$ (versus control); ##$p < 0.01$ (versus model). Scale bar, 50 μm or 100 μm.

Ethics. The use of animals in this study was approved by the animal centre of the Shanghai University of Traditional Chinese Medicine under the permit SYXK (HU) 2014-0008.

Data accessibility. All data of DEGs supporting this paper have been uploaded as electronic supplementary material.

Authors' contributions. C.C. and Z.Y. made substantial contributions to the conception of the project, F.D. is responsible for all molecular experiments and histological experiments, L.S. is responsible for cell culture and lipid peroxidative injury model construction, B.W. and T.L. are responsible for data analysis, and J.C. is responsible for statistical analysis of data. All authors were involved in drafting and revising the article, and approving the article.

Competing interests. The authors declare that they have no financial or non-financial competing interests.

Funding. The present study was supported by grants from the Natural Science Foundation of China (programme no. 81704129) and from the Shanghai Health Bureau Youth Fund (programme nos. 201640232 and ZY (2018-2020)-CCCX-4004).

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
