## [Reviewer comments · Open Biology]

Review History

RSOB-19-0141.R0 (Original submission)

Review form: Reviewer 1

Recommendation

Major revision is needed (please make suggestions in comments)

Do you have any ethical concerns with this paper?

No

Comments to the Author

In the current manuscript by Dou et al. entitled "Identification of a novel regulatory pathway for PPAR α by RNA-seq characterization of the endothelial cell lipid peroxidative injury transcriptome", the authors identified differentially expression genes after endothelial cell injury in HUVEC cells via RNA-seq. The predicted PPIs network suggested that CCL2 may play a central role after endothelial cell injury. Investigation into CCL2 and PPAR α revealed that

PPAR α activated downstream CCL2 expression in HUVEC cells and in ApoE $^{-/-}$ mice model. By comparison of proliferation, migration and angiogenesis in HUVEC cells with modified PPAR α activity and CCL2 expression, they found that the function of PPAR α during the repair of endothelial cell injury is dependent on CCL2. Thus, this manuscript uncovered the role of PPAR α -CCL2 during the repair of lipid injury. However, the authors could strengthen their finding in ApoE $^{-/-}$ mice model. It seems that the “damage stage” and “repair stage” after endothelial cell injury was confused (see below). In addition, the experimental rationales should be improved. These points prevent me from recommending the manuscript for publication unless the following concerns could be carefully addressed.

Major points:

1. Page 11, for readers' convenience, the authors should briefly introduce the aim of the study and the workflow (cell/animal model, sample collection and data analysis) at the beginning of the “Results” section, although part of these information has been provided in “Materials and Methods”.
2. Figure 1, the authors used HUVEC cells treated with 200 μ g/mL of oxLDL for 24 hours as cellular model of endothelial injury. The time point after injury is important for cellular response (acute, chronic response/repair stages) to injury/stress. Have the authors optimized with different time points and oxLDL doses, and verified the in vitro vascular endothelial cell injury model?
3. Figure 1, the authors have successfully established animal model (ApoE $^{-/-}$ mice fed with a high-fat diet), they could cross validate their RNA-seq results from cell model vs animal model.
4. Figure 2, the rationale is not clear from RNA-Seq to PPIs network, and back to validation of mRNA expression. The authors should also consider subcellular localization to improve the reliability of the predicted PPIs, e.g. Figure 2 predicted that CCL2 (extracellular or membrane) and PPAR α (nuclear) are interacted, they are assumed to be co-localized.
5. Figure 3, the rationale is not clear to choose CCL2 and PPAR α to “speculate CCL2 and PPAR α play critical roles in the injury repair caused by lipid peroxidation” and ignore PDGFB, NOX4, VWF while all of them shared similar expression pattern after lipid injury.
6. Figure 4B, RT-qPCR of CCL2 is suggested to distinguish whether CCL2 is regulated by PPAR α at transcription or post-transcription level. Prediction of PPAR α consensus binding site in the promoter region of CCL2 gene is suggested to distinguish whether CCL2 is regulated by PPAR α directly or indirectly.
7. Figure 4C and page 13, the authors should briefly introduce animal models used here (ApoE $^{-/-}$ normal diet mice as control group and ApoE $^{-/-}$ high-fat diet mice as model group), although mentioned in “Materials and Methods”. Have the authors extracted ventricular outflow tract tissues to quantify CCL2 expression among control, model, PPAR α agonist and PPAR α antagonist groups?
8. Page 13 Line 4, the authors mentioned that “we found a small amount of CCL2 expression in the control group and the model group”, this indicated that there are no up-regulation of CCL2 in mice model of lipid injury, while data from cell model (oxLDL for 24 hrs, without PPAR α agonist) showed induction of CCL2 and PPAR α . Could the author explain the discrepancy is due to different model or different stages (damage, repair) after lipid injury? The authors should also investigate PPAR α expression among these animal models because the activation of PPAR α maybe a hallmark of repair stage.
9. Figure 6, the CCL2 expression level should be included in parallel with the effect of PPAR α on proliferation, migration and angiogenesis. The control group (without oxLDL) should also be included.

Minor points:

1. Pages 9 line 16, “Sections were washed, mounted, and examined using a light microscope”, please check whether “light microscope” is correct for examination of immunofluorescence staining.

2. Figure 4C, the labeling of “control” and “model” can not distinguish animal model used here from cell model used elsewhere. The authors could label with “ApoE^{-/-} mice control” and “ApoE^{-/-} mice model”.
3. Table 1 is not necessary. The authors should provide the details of RNA-seq results (related to Figure 1) including names of total 311 genes and fold changes into Supplemental Table 1.

Decision letter (RSOB-19-0141.R0)

19-Aug-2019

Dear Professor Chen,

We are writing to inform you that the Editor has reached a decision on your manuscript RSOB-19-0141 entitled "Identification of a novel regulatory pathway for PPAR α by RNA-seq characterization of the endothelial cell lipid peroxidative injury transcriptome", submitted to Open Biology.

As you will see from the reviewer's comments below, there are a number of criticisms that prevent us from accepting your manuscript at this stage. The reviewer suggests, however, that a revised version could be acceptable, if you are able to address their concerns. If you think that you can deal satisfactorily with the reviewer's suggestions, we would be pleased to consider a revised manuscript.

The revision will be re-reviewed, where possible, by the original referee(s). As such, please submit the revised version of your manuscript within six weeks. If you do not think you will be able to meet this date please let us know immediately.

To revise your manuscript, log into <https://mc.manuscriptcentral.com//rsob> and enter your Author Centre, where you will find your manuscript title listed under "Manuscripts with Decisions." Under "Actions," click on "Create a Revision." Your manuscript number has been appended to denote a revision.

When submitting your revised manuscript, please respond to the comments made by the referee(s) and upload a file "Response to Referees" in "Section 6 - File Upload". You can use this to document any changes you make to the original manuscript. In order to expedite the processing of the revised manuscript, please be as specific as possible in your response to the referee(s).

Please see our detailed instructions for revision requirements
<https://royalsociety.org/journals/authors/author-guidelines/>

Sincerely,
The Open Biology Team
<mailto:openbiology@royalsociety.org>

Reviewer(s)' Comments to Author(s):

Referee: 1

Comments to the Author(s)

In the current manuscript by Dou et al. entitled "Identification of a novel regulatory pathway for PPAR α by RNA-seq characterization of the endothelial cell lipid peroxidative injury transcriptome", the authors identified differentially expression genes after endothelial cell injury in HUVEC cells via RNA-seq. The predicted PPIs network suggested that CCL2 may play a central role after endothelial cell injury. Investigation into CCL2 and PPAR α revealed that PPAR α activated downstream CCL2 expression in HUVEC cells and in ApoE^{-/-} mice model. By comparison of proliferation, migration and angiogenesis in HUVEC cells with modified PPAR α activity and CCL2 expression, they found that the function of PPAR α during the repair of endothelial cell injury is dependent on CCL2. Thus, this manuscript uncovered the role of PPAR α -CCL2 during the repair of lipid injury. However, the authors could strengthen their finding in ApoE^{-/-} mice model. It seems that the "damage stage" and "repair stage" after endothelial cell injury was confused (see below). In addition, the experimental rationales should be improved. These points prevent me from recommending the manuscript for publication unless the following concerns could be carefully addressed.

Major points:

1. Page 11, for readers' convenience, the authors should briefly introduce the aim of the study and the workflow (cell/animal model, sample collection and data analysis) at the beginning of the "Results" section, although part of these information has been provided in "Materials and Methods".
2. Figure 1, the authors used HUVEC cells treated with 200 μ g/mL of oxLDL for 24 hours as cellular model of endothelial injury. The time point after injury is important for cellular response (acute, chronic response/repair stages) to injury/stress. Have the authors optimized with different time points and oxLDL doses, and verified the in vitro vascular endothelial cell injury model?
3. Figure 1, the authors have successfully established animal model (ApoE^{-/-} mice fed with a high-fat diet), they could cross validate their RNA-seq results from cell model vs animal model.
4. Figure 2, the rationale is not clear from RNA-Seq to PPIs network, and back to validation of mRNA expression. The authors should also consider subcellular localization to improve the reliability of the predicted PPIs, e.g. Figure 2 predicted that CCL2 (extracellular or membrane) and PPAR α (nuclear) are interacted, they are assumed to be co-localized.
5. Figure 3, the rationale is not clear to choose CCL2 and PPAR α to "speculate CCL2 and PPAR α play critical roles in the injury repair caused by lipid peroxidation" and ignore PDGFB, NOX4, VWF while all of them shared similar expression pattern after lipid injury.
6. Figure 4B, RT-qPCR of CCL2 is suggested to distinguish whether CCL2 is regulated by PPAR α at transcription or post-transcription level. Prediction of PPAR α consensus binding site in the promoter region of CCL2 gene is suggested to distinguish whether CCL2 is regulated by PPAR α directly or indirectly.
7. Figure 4C and page 13, the authors should briefly introduce animal models used here (ApoE^{-/-} normal diet mice as control group and ApoE^{-/-} high-fat diet mice as model group), although mentioned in "Materials and Methods". Have the authors extracted ventricular outflow tract tissues to quantify CCL2 expression among control, model, PPAR α agonist and PPAR α antagonist groups?
8. Page 13 Line 4, the authors mentioned that "we found a small amount of CCL2 expression in the control group and the model group", this indicated that there are no up-regulation of CCL2 in mice model of lipid injury, while data from cell model (oxLDL for 24 hrs, without PPAR α agonist) showed induction of CCL2 and PPAR α . Could the author explain the discrepancy is

due to different model or different stages (damage, repair) after lipid injury? The authors should also investigate PPAR α expression among these animal models because the activation of PPAR α maybe a hallmark of repair stage.

9. Figure 6, the CCL2 expression level should be included in parallel with the effect of PPAR α on proliferation, migration and angiogenesis. The control group (without oxLDL) should also be included.

Minor points:

1. Pages 9 line 16, "Sections were washed, mounted, and examined using a light microscope", please check whether "light microscope" is correct for examination of immunofluorescence staining.
2. Figure 4C, the labeling of "control" and "model" can not distinguish animal model used here from cell model used elsewhere. The authors could label with "ApoE^{-/-} mice control" and "ApoE^{-/-} mice model".
3. Table 1 is not necessary. The authors should provide the details of RNA-seq results (related to Figure 1) including names of total 311 genes and fold changes into Supplemental Table 1.

Author's Response to Decision Letter for (RSOB-19-0141.R0)

See Appendix A.

RSOB-19-0141.R1 (Revision)

Review form: Reviewer 1

Recommendation

Accept with minor revision (please list in comments)

Do you have any ethical concerns with this paper?

No

Comments to the Author

The authors have substantially improved the revised version of this manuscript and have sufficiently addressed my previous concerns.

I have now only a few minor concerns:

1. Page 13, lines 1-18 (the paragraph before section 3.1) seem a summary for this study and are unusual, weird in regards to paper format. In fact, a brief introduction of the background of transcriptome analysis in this study is sufficient for understanding by the readers. I would suggest moving lines 1-7 to the beginning of section 3.1, while lines 8-18 could either move to the end of section 1 (Introduction) or be deleted.
2. To avoid of confusion, I would suggest adding the following (or something like that) at the end of section 3.1: The interactions between PPI-predicted proteins include direct (physical) and indirect (functional) associations.
3. Sections 3.2 and 3.3 could be combined. Section 3.5 could be split into 2 sections: one is related to Fig. 4 (PPAR α regulates CCL2 expression) and another is related to Fig. 5 and 6 (PPAR α

participates in the repair of cell injury induced by lipid peroxidation through regulating CCL2 expression).

Decision letter (RSOB-19-0141.R1)

11-Nov-2019

Dear Professor Chen

We are pleased to inform you that your manuscript RSOB-19-0141.R1 entitled "Identification of a novel regulatory pathway for PPAR α by RNA-seq characterization of the endothelial cell lipid peroxidative injury transcriptome" has been accepted by the Editor for publication in Open Biology. The reviewer(s) have recommended publication, but also suggest some minor revisions to your manuscript. Therefore, we invite you to respond to the reviewer(s)' comments and revise your manuscript.

Please submit the revised version of your manuscript within 7 days. If you do not think you will be able to meet this date please let us know immediately and we can extend this deadline for you.

- 1) A text file of the manuscript (doc, txt, rtf or tex), including the references, tables (including captions) and figure captions. Please remove any tracked changes from the text before submission. PDF files are not an accepted format for the "Main Document".
- 2) A separate electronic file of each figure (tiff, EPS or print-quality PDF preferred). The format should be produced directly from original creation package, or original software format. Please note that PowerPoint files are not accepted.
- 3) Electronic supplementary material: this should be contained in a separate file from the main text and meet our ESM criteria (see <http://royalsocietypublishing.org/instructions-authors#question5>). All supplementary materials accompanying an accepted article will be

treated as in their final form. They will be published alongside the paper on the journal website and posted on the online figshare repository. Files on figshare will be made available approximately one week before the accompanying article so that the supplementary material can be attributed a unique DOI.

Online supplementary material will also carry the title and description provided during submission, so please ensure these are accurate and informative. Note that the Royal Society will not edit or typeset supplementary material and it will be hosted as provided. Please ensure that the supplementary material includes the paper details (authors, title, journal name, article DOI). Your article DOI will be 10.1098/rsob.2016[last 4 digits of e.g. 10.1098/rsob.20160049].

4) A media summary: a short non-technical summary (up to 100 words) of the key findings/importance of your manuscript. Please try to write in simple English, avoid jargon, explain the importance of the topic, outline the main implications and describe why this topic is newsworthy.

Images

Data-Sharing

It is a condition of publication that data supporting your paper are made available. Data should be made available either in the electronic supplementary material or through an appropriate repository. Details of how to access data should be included in your paper. Please see <http://royalsocietypublishing.org/site/authors/policy.xhtml#question6> for more details.

Data accessibility section

Sincerely,

The Open Biology Team
<mailto:openbiology@royalsociety.org>

Reviewer(s)' Comments to Author:

Referee: 1

Comments to the Author(s)

The authors have substantially improved the revised version of this manuscript and have sufficiently addressed my previous concerns.

I have now only a few minor concerns:

1. Page 13, lines 1-18 (the paragraph before section 3.1) seem a summary for this study and are unusual, weird in regards to paper format. In fact, a brief introduction of the background of transcriptome analysis in this study is sufficient for understanding by the readers. I would suggest moving lines 1-7 to the beginning of section 3.1, while lines 8-18 could either move to the end of section 1 (Introduction) or be deleted.
2. To avoid of confusion, I would suggest adding the following (or something like that) at the end of section 3.1: The interactions between PPI-predicted proteins include direct (physical) and indirect (functional) associations.
3. Sections 3.2 and 3.3 could be combined. Section 3.5 could be split into 2 sections: one is related to Fig. 4 (PPAR α regulates CCL2 expression) and another is related to Fig. 5 and 6 (PPAR α participates in the repair of cell injury induced by lipid peroxidation through regulating CCL2 expression).

Author's Response to Decision Letter for (RSOB-19-0141.R1)

See Appendix B.

Decision letter (RSOB-19-0141.R2)

15-Nov-2019

Dear Professor Chen

We are pleased to inform you that your manuscript entitled "Identification of a novel regulatory pathway for PPAR α by RNA-seq characterization of the endothelial cell lipid peroxidative injury transcriptome" has been accepted by the Editor for publication in Open Biology.

Article processing charge

Please note that the article processing charge is immediately payable. A separate email will be sent out shortly to confirm the charge due. The preferred payment method is by credit card; however, other payment options are available.

Sincerely,

The Open Biology Team
mailto:openbiology@royalsociety.org

Appendix A

Response to Referees

Referee: 1

Comments to the Author(s)

In the current manuscript by Dou et al. entitled “Identification of a novel regulatory pathway for PPAR α by RNA-seq characterization of the endothelial cell lipid peroxidative injury transcriptome”, the authors identified differentially expression genes after endothelial cell injury in HUVEC cells via RNA-seq. The predicted PPIs network suggested that CCL2 may play a central role after endothelial cell injury. Investigation into CCL2 and PPAR α revealed that PPAR α activated downstream CCL2 expression in HUVEC cells and in ApoE $^{-/-}$ mice model. By comparison of proliferation, migration and angiogenesis in HUVEC cells with modified PPAR α activity and CCL2 expression, they found that the function of PPAR α during the repair of endothelial cell injury is dependent on CCL2. Thus, this manuscript uncovered the role of PPAR α -CCL2 during the repair of lipid injury. However, the authors could strengthen their finding in ApoE $^{-/-}$ mice model. It seems that the “damage stage” and “repair stage” after endothelial cell injury was confused (see below). In addition, the experimental rationales should be improved. These points prevent me from recommending the manuscript for publication unless the following concerns could be carefully addressed.

ANSWER: Dear editor and reviewer, thank you very much for giving us the opportunity to revise the manuscript, we will provide point-by-piont response to the comments and resubmit the improved manuscript online.

Major points:

1. Page 11, for readers’ convenience, the authors should briefly introduce the aim of the study and the workflow (cell/animal model, sample collection and data analysis) at the beginning of the “Results” section, although part of these information has been provided in “Materials and Methods”.

Answer 1: Dear reviewer, thanks for your advice firstly, we have added brief

introduction of the aim of the study and the workflow at the beginning of the “Results” section in revised manuscript, please check.

2. Figure 1, the authors used HUVEC cells treated with 200 µg/mL of oxLDL for 24 hours as cellular model of endothelial injury. The time point after injury is important for cellular response (acute, chronic response/repair stages) to injury/stress. Have the authors optimized with different time points and oxLDL doses, and verified the in vitro vascular endothelial cell injury model?

Answer 2: Dear reviewer, thank you very much for your professional advice, ox-LDL-induced endothelial cell injury model is an simulated atherosclerotic injury model *in vitro* we have been using, in order to build the cell model, different concentrations of ox-LDL (10 µg/mL, 50 µg/mL, 100 µg/mL, 200 µg/mL and 300 µg/mL) were added to cultured HUVECs *in vitro*, at different time points (6 hr, 12 hr, 24 hr and 48 hr), we used CCK8, TUNEL, Oil red O assay to confirm the cell viability and proliferation, apoptosis and damage, results showed that 10 µg/mL and 50 µg/mL ox-LDL did not affect the cell viability and proliferation, apoptosis and damage at all time points, 100 µg/mL ox-LDL could promote cell viability and proliferation at all time points, 200 µg/mL ox-LDL inhibited approximately 20% of cell viability and proliferation, resulted in approximately 20% cell apoptosis, while 80% of cell showed positive oil red O staining at 24 hr only, we did not detect the difference between injury group and control group at 6 hr and 12 hr, at 48 hr, there was no significant difference in cell viability, proliferation and apoptosis between the ox-LDL group and the control group. 300 µg/mL ox-LDL resulted in more than 50% cell apoptosis. Therefore, we chosen 200 µg/mL as our final concentration.

3. Figure 1, the authors have successfully established animal model (ApoE^{-/-} mice fed with a high-fat diet), they could cross validate their RNA-seq results from cell model vs animal model.

Answer 3: Dear reviewer, thank you very much for your advice, since the cells we are using is human umbilical vein endothelial cells, although transcripts of human and

mouse have high homology, different transcripts may transcribe proteins with different functional sites. Therefore, we did not cross validate their RNA-seq results from cell model vs animal model. However, we believe you gave us a good suggestion, we will use a mouse vascular endothelial cell to make a damage model, and compare the RNA-seq results with the results of *ApoE*^{-/-} mice to obtain a more reliable conclusion.

4. Figure 2, the rationale is not clear from RNA-Seq to PPIs network, and back to validation of mRNA expression. The authors should also consider subcellular localization to improve the reliability of the predicted PPIs, e.g. Figure 2 predicted that CCL2 (extracellular or membrane) and PPAR α (nuclear) are interacted, they are assumed to be co-localized.

Answer 4: Dear reviewer, thanks for your professional advice firstly. As you said, there is no direct connection between RNA-seq and PPI network. RNA-seq is only used to detect the expression and changes of a molecular at the RNA level, changes of protein level are not consistent with changes in RNA expression, and PPI network prediction is only a speculation of the relationship between proteins according to the results of RNA-seq. The interactions between PPI-predicted proteins include direct physical interactions and indirect correlations. CCL2 is a chemokine, and chemokines mostly perform its function as secreted proteins, PPAR α is a transcription factor, it exerts regulatory functions by binding to the promoter region of the target protein, when the PPI network predicts that there is an interaction between CCL2 and PPAR α , we speculate that PPAR α binds to the promoter region of CCL2 and regulates the expression of CCL2. Therefore, the PPI network is just a tool we use to speculate whether there is interaction between proteins, and we need verified experiments to proving the authenticity of this speculation.

5. Figure 3, the rationale is not clear to choose CCL2 and PPAR α to “speculate CCL2 and PPAR α play critical roles in the injury repair caused by lipid peroxidation” and ignore PDGFB, NOX4, VWF while all of them shared similar expression pattern after

lipid injury.

Answer 5: Dear reviewer, thank you very much for your valuable comments on the issues in our manuscript. Because the interactions between PPI-predicted proteins include direct physical interactions and indirect correlations, by analyzing each protein-related function, PDGF β and VWF can be used as marker proteins of endothelial cells, and as the main protease producing reactive oxygen species, NOX4 is usually in a highly active state after endothelial cell injury, their interaction with the chemokine CCL2 may not be a direct regulatory relationship, and PPARA, as a transcription factor, might directly bind to the promoter region of CCL2 to regulate its expression. Therefore, we firstly selected PPARA and CCL2 as research targets. Based on your suggestions, we will continue to study the relationship between NOX4, PDGF β , VWF and CCL2, respectively.

6. Figure 4B, RT-qPCR of CCL2 is suggested to distinguish whether CCL2 is regulated by PPAR α at transcription or post-transcription level. Prediction of PPAR α consensus binding site in the promoter region of CCL2 gene is suggested to distinguish whether CCL2 is regulated by PPAR α directly or indirectly.

Answer 6: Dear reviewer, thank you very much for your professional advice. RT-qPCR assay of CCL2 has been performed and proved that CCL2 was regulated by PPAR α at transcription level, and luciferase assay confirmed that PPAR α could bind to the promoter region of CCL2 gene to promote CCL2 expression. These results and experiment methods (2.6 Luciferase reporter assay) have been added to figure 4 and to manuscript, please check.

7. Figure 4C and page 13, the authors should briefly introduce animal models used here (ApoE $^{-/-}$ normal diet mice as control group and ApoE $^{-/-}$ high-fat diet mice as model group), although mentioned in “Materials and Methods”. Have the authors extracted ventricular outflow tract tissues to quantify CCL2 expression among control, model, PPAR α agonist and PPAR α antagonist groups?

Answer 7: Dear reviewer, thanks for your professional advice. We have gave a brief

introduction of animal model in page 13 and figure 4, and we also examined the expression of CCL2 of the left ventricular outflow tract tissues in *Apoe*^{-/-} mice, please check.

8. Page 13 Line 4, the authors mentioned that “we found a small amount of CCL2 expression in the control group and the model group”, this indicated that there are no up-regulation of CCL2 in mice model of lipid injury, while data from cell model (oxLDL for 24 hrs, without PPAR α agonist) showed induction of CCL2 and PPAR α . Could the author explain the discrepancy is due to different model or different stages (damage, repair) after lipid injury? The authors should also investigate PPAR α expression among these animal models because the activation of PPAR α maybe a hallmark of repair stage.

Answer 8: Dear reviewer, thank you for pointing out the flaws in the manuscript. The description of this sentence “we found a small amount of CCL2 expression in the control group and the model group” may lead to some misunderstanding, the intention of this sentence is to show that the expression of CCL2 in the *Apoe*^{-/-} mice with normal diet and high fat diet is relatively small compared with the *Apoe*^{-/-} mice using PPAR α agonists. We did not make a statistical comparison of the expression levels of CCL2 in the control group and the model group in immunofluorescence staining, the results of detecting CCL2 and PPAR α protein expression of the left ventricular outflow tract tissues showed that the expression of the two proteins have no significant difference between the control group and the model group, we believe that there are some differences of data from cell model and mice model are reasonable, because *Apoe*^{-/-} mice are an animal model of spontaneous atherosclerosis, *Apoe*^{-/-} mice develop atherosclerotic plaques from 2 months of age, with or without a high-fat diet, the high-fat diet is only an accelerator to the plaque, the aortic tissue we detected comes from 5-month-old *Apoe*^{-/-} mice, which had formed stable plaques with or without a high-fat diet, that is the reason that we did not detect expressional differences of CCL2 and PPAR α between the control group and the model group. While data from cell model (oxLDL for 24 hrs) showed induction of CCL2 and

PPAR α because of normal cells without any treatment as the control group, CCL2 and PPAR α expression level were induced by ox-LDL. We have examined the PPAR α expression among these animal models, please check.

9. Figure 6, the CCL2 expression level should be included in parallel with the effect of PPAR α on proliferation, migration and angiogenesis. The control group (without oxLDL) should also be included.

Answer 9: Dear reviewer, thanks for your professional advice firstly. We have detected the CCL2 expression level after adding PPAR α agonist to the stable CCL2 knockdown HUVECs, in order to investigate whether apoptosis, proliferation, migration and angiogenesis of PPAR α were achieved by regulating CCL2. And the control group also added in figure 6, please check.

Minor points:

1. Pages 9 line 16, “Sections were washed, mounted, and examined using a light microscope”, please check whether “light microscope” is correct for examination of immunofluorescence staining.

Answer 1: Dear reviewer, thank you very much for pointing out our writing mistakes, the microscope (CX43 Biological Microscope, Olympus, Japan) we use is fluorescence microscope, so we have corrected the “light microscope” to “fluorescence microscope” in the manuscript, thanks again.

2. Figure 4C, the labeling of “control” and “model” can not distinguish animal model used here from cell model used elsewhere. The authors could label with “ApoE $^{-/-}$ mice control” and “ApoE $^{-/-}$ mice model”.

Answer 2: Dear reviewer, based on your suggestions, we have label with “ApoE $^{-/-}$ mice control” and “ApoE $^{-/-}$ mice model” in figure 4, please check.

3. Table 1 is not necessary. The authors should provide the details of RNA-seq results (related to Figure 1) including names of total 311 genes and fold changes into

Supplemental Table 1.

Answer 3: Dear reviewer, thanks you very much for you advice, we have added the details of RNA-seq results including names of total 311 genes and fold changes into Supplemental Table 1 named “Differentially Expressed Genes (DEGs)” according to your suggestions, please check.

Appendix B

Response to Referees

Comments to the Author(s)

The authors have substantially improved the revised version of this manuscript and have sufficiently addressed my previous concerns.

ANSWER: Dear editor and reviewer, thank you very much for giving us the opportunity to revise and publish the manuscript, we will provide point-by-point response to the comments and resubmit the improved manuscript online.

I have now only a few minor concerns:

1. Page 13, lines 1-18 (the paragraph before section 3.1) seem a summary for this study and are unusual, weird in regards to paper format. In fact, a brief introduction of the background of transcriptome analysis in this study is sufficient for understanding by the readers. I would suggest moving lines 1-7 to the beginning of section 3.1, while lines 8-18 could either move to the end of section 1 (Introduction) or be deleted.

Answer 1: Dear reviewer, thank you very much for your professional advice, we have retained line 1-7 at the beginning of section 3.1, while deleted line 8-18 because these contents are similarly described in the introduction.

2. To avoid of confusion, I would suggest adding the following (or something like that) at the end of section 3.1: The interactions between PPI-predicted proteins include direct (physical) and indirect (functional) associations.

Answer 2: Dear reviewer, thank you very much for making such meticulous revisions to the manuscript, according to your advice, we have added “The interactions between PPI-predicted proteins include direct (physical) and indirect (functional) associations” at the end of section 3.1, thanks again.

3. Sections 3.2 and 3.3 could be combined. Section 3.5 could be split into 2 sections: one is related to Fig. 4 (PPAR α regulates CCL2 expression) and another is related to Fig. 5 and 6 (PPAR α participates in the repair of cell injury induced by lipid peroxidation through regulating CCL2 expression).

Answer 3: Dear reviewer, thank you very much for your professional advice, we have combined section 3.2 and section 3.3, and splitted section 3.5 into 2 sections according to your comments. Please check.